

# Transcriptome analysis and exploration of genes involved in the biosynthesis of secoiridoids in *Gentiana rhodantha*

Ting Zhang[1,2], Miaomiao Wang[1], Zhaoju Li[1], Xien Wu[1] and Xiaoli Liu[1,2]

[1] College of Chinese Material Medica, Yunnan University of Chinese Medicine, Kunming, Yunnan, China
[2] Medicine Yunnan Provincial Key Laboratory of Molecular Biology for Sino Medicine, Yunnan University of Chinese Medicine, Kunming, Yunnan, China

## ABSTRACT

*Gentiana rhodantha* is a medicinally important perennial herb used as traditional Chinese and ethnic medicines. Secoiridoids are one of the major bioactive compounds in *G. rhodantha*. To better understand the secoiridoid biosynthesis pathway, we generated transcriptome sequences from four organs (root, leaf, stem and flower), followed by the *de novo* sequence assembly. We verified *8-HGO* (*8-hydroxygeraniol oxidoreductase*), which may encode key enzymes of the secoiridoid biosynthesis by qRT-PCR. The mangiferin, swertiamarin and loganic acid contents in root, stem, leaf, and flower were determined by HPLC. The results showed that there were 47,871 unigenes with an average length of 1,107.38 bp. Among them, 1,422 unigenes were involved in 25 standard secondary metabolism-related pathways in the KEGG database. Furthermore, we found that 1,005 unigenes can be divided into 66 transcription factor (TF) families, with no family members exhibiting significant organ-specificity. There were 54 unigenes in *G. rhodantha* that encoded 17 key enzymes of the secoiridoid biosynthetic pathway. The qRT-PCR of the *8-HGO* and HPLC results showed that the relative expression and the mangiferin, swertiamarin, and loganic acid contents of the aerial parts were higher than in the root. Six types of SSR were identified by SSR analysis of unigenes: mono-nucleoside repeat SSR, di-nucleoside repeat SSR, tri-nucleoside repeat SSR, tetra-nucleoside repeat SSR, penta-nucleoside repeat SSR, and hexa-nucleoside repeat SSR. This report not only enriches the *Gentiana* transcriptome database but helps further study the function and regulation of active component biosynthesis of *G. rhodantha*.

## INTRODUCTION

The herbaceous *Gentiana* genus (family Gentianaceae) comprises about 500 species worldwide, and is widely distributed in the temperate and tropical alpine regions of the northern hemisphere including Europe, Asia, northern Australia, New Zealand, North America, reaches Cape Horn along the Andes and northern Africa. There are approximately 247 species in China, which are mainly distributed in the southwest mountainous area. *Gentiana* has multiple pharmacological effects, including hepatoprotective, anti-inflammatory, antipyretic, *etc* (*Editorial Committee of Chinese Flora, 1988*; *Dong et al.,*

Corresponding author
Xiaoli Liu, kmxunzi@aliyun.com

*2017*). Many *Gentiana* species, including *G. scabra*, *G. rigescens*, *G. macrophylla*, and *G. rhodantha* have been recorded in the Chinese Pharmacopoeia. They are all perennially erect herbs with bluish-purple flowers. *G. scabra* and *G. rigescens* are officially listed in the Chinese Pharmacopoeia under the name *Gentianae radix* et *rhizoma* (Longdan and Jianlongdan, respectively, in Chinese). They have been used for jaundice, eczema, and acute conjunctivitis. Moreover, *G. macrophylla* is perennial and officially listed as *Gentianae macrophyllae* radix (Qinjiao in Chinese) for rheumatic arthralgia, poplexy and hemiplegia. The dried whole herbs of *G. rhodantha* are officially listed in the Chinese Pharmacopoeia as *Gentianae rhodanthae* herb (Honghualongdan in Chinese) for jaundice, detoxification, and relieving cough (*Chinese Pharmacopoeia Commission, 2020*). It is not only used as traditional Chinese medicine but also as ethno-medicine in China. It is widely used in ethnic minorities, including Miao, Buyi, Bai, Yao, and Tujia (*Luo, 1990*; *Mo et al., 2003*; *Zhao, 2005*; *Jiang, 2015*). To date, many Chinese medicines related to *G. rhodantha* have been developed. Among them, the Feilike mixture treats coughing up sputum, poor breathing, acute and chronic bronchitis, emphysema, and other symptoms (*Jin, 2012*). The Kangfuling tablet is used for treating gynecological diseases, including cervicitis, vaginitis, menstrual irregularities, red vaginal discharge, dysmenorrhea, adnexitis, *etc* (*Li, 2013*). Furthermore, the Lianlong capsule helps in reducing swelling and loosening of knots, along with treatment of thyroid tumors, liver cancer, and other malignant tumors (*Yang et al., 2016*). In recent years, *G. rhodantha* wild resources have been declining due to low seed germination rates, poor reproductive ability, and overexploitation (*Sun et al., 2016*; *Shen et al., 2017*). Fortunately, tissue culture and a rapid propagation system had been established, making it possible to artificially produce *G. rhodantha* and regulate its quality (*Zhong et al., 2021*).

RNA sequencing (RNA-seq) is a powerful technology for genome-wide analysis of RNA transcripts (*Grabherr et al., 2011*). High-quality transcriptome data not only mine genomic resources, but also facilitate genetic and molecular breeding approaches for metabolic regulation in medicinal plants (*Qi, Liu & Rong, 2011*). Flavonoids, secoiridoids, and phenolic acids were the main active components in *G. rhodantha* (*Xu et al., 2011*; *Ma, Fuzzati & Wolfender, 1996*; *Xu et al., 2008*; *Yao, Wu & Chou, 2015*). Mangiferin is a carbon glycoside of tetrahydroxypyrone, which belongs to the diphenylpyrone group of compounds. It has special properties and has attracted extensive interest as a therapeutic metabolite (*Fan et al., 2017*; *Wang et al., 2022*). Secoiridoids, like swertiamarin, are also the main active components of *G. rhodantha*, which have rich pharmacological effects, including analgesia, hypotension, and osteoblasts proliferation (*Chen, 2009*; *Xu et al., 2008*; *Sun et al., 2008*).

The transcriptomic studies for determining the genes involved in the secoiridoid biosynthetic pathway were performed for multiple *Gentiana* species, *including G. rigescens*, *G.crassicaulis, and G. waltonii* (*Kang et al., 2021*; *Zhang et al., 2015*; *Ni et al., 2019*). However, only the chloroplast genome *G. rhodantha* has been reported (*Hu & Zhang, 2021*; *Ling, 2020*). The mangiferin and secoiridoid biosynthetic pathway is still unknown in *G. rhodantha*. We first sequenced the transcripts of the root, stem, leaf, and flower collected at the full bloom stage for *G. rhodantha* using the IIlumina Hiseq 6000 high-throughput
platform, and then deciphered all the candidate genes and putative transcription factors involved in the mangiferin and secoiridoid biosynthetic pathway. Although mangiferin is one of the main active components of *G. rhodantha*, the mangiferin biosynthetic pathway is unavailable in the KEGG official website to date. Therefore, this study focused on the analysis and identification of the putative genes involved in secoiridoid biosynthesis, aiming to provide useful insights into their further quality regulation. Additionally, numerous simple sequence repeats (SSR) markers were found, which will facilitate the marker-assisted breeding of *G. rhodantha*.

## MATERIALS AND METHODS

*G. rhodantha* used in this experiment, was collected from the Yiduoyun village, Kunming, Yunnan, China (25°00′5.62″N, 102°58′46.35″E, 1,068 m). The taxonomic identities of the voucher specimens were identified by the corresponding author. Natural wild *G. rhodantha* was sampled at the full bloom stage. Fresh root, stem, leaf, and flower were collected from three plants having relatively consistent growth and synchronized development (Figs. 1A, 1B). One half was quickly frozen in liquid nitrogen and stored at −80 °C, while the other half was dried to constant weight at 55 °C and used for determining the mangiferin, swertiamarin, and loganic acid contents.

### RNA extraction and cDNA library preparation and RNA-Seq analysis

RNA was extracted from the root, leaf, stem, and flower. First, 1 μg RNA per sample was used for the RNA sample preparations. Sequencing libraries were generated using the NEBNext®Ultra™ RNA Library Prep Kit for Illumina® (New England Biolabs, Ipswich, MA, USA) following the manufacturer's recommendations and index codes were added to attribute sequences to each sample. Twelve libraries were obtained from three biological replicates per organ. Illumina Novaseq 6,000 sequencing was performed after the library was qualified. The average sequencing depth (count* 150/gene_len) was approximately 456.

A power analysis was performed by RNASeqpower in the R package (*Hart et al., 2013*). The statistical power of this experimental design, calculated in RNASeqpower, was 0.9347033 (depth = 456, $n = 3$, CV = 0.24, effect = 2, alpha = 0.05).

### *De novo* assembly and sequence processing

First, the raw data (raw reads) of the fastq format were processed through the in-house Perl scripts. Filter joint, low quality, sequences with N bases, low quality bases ($Q < 20$) were removed to get high-quality clean data (*Bolger, Marc & Bjoern, 2014*). Firstly, Breaking sequencing reads into short segments (K-mer) *via* TRINITY (https://github.com/trinityrnaseq/trinityrnaseq/wiki) under the parameter of (–min_contig_length 200, –group_pairs_distance 500). These small fragments were then extended into longer segments (contigs). Overlaps between these fragments were used to get a collection of fragments (component). Finally, the De Brujin graph method and sequencing read information were used to identify transcript sequences in each fragment collection. The RSeQC (RNA-seq data QC) software was used to remove the redundant sequences in the

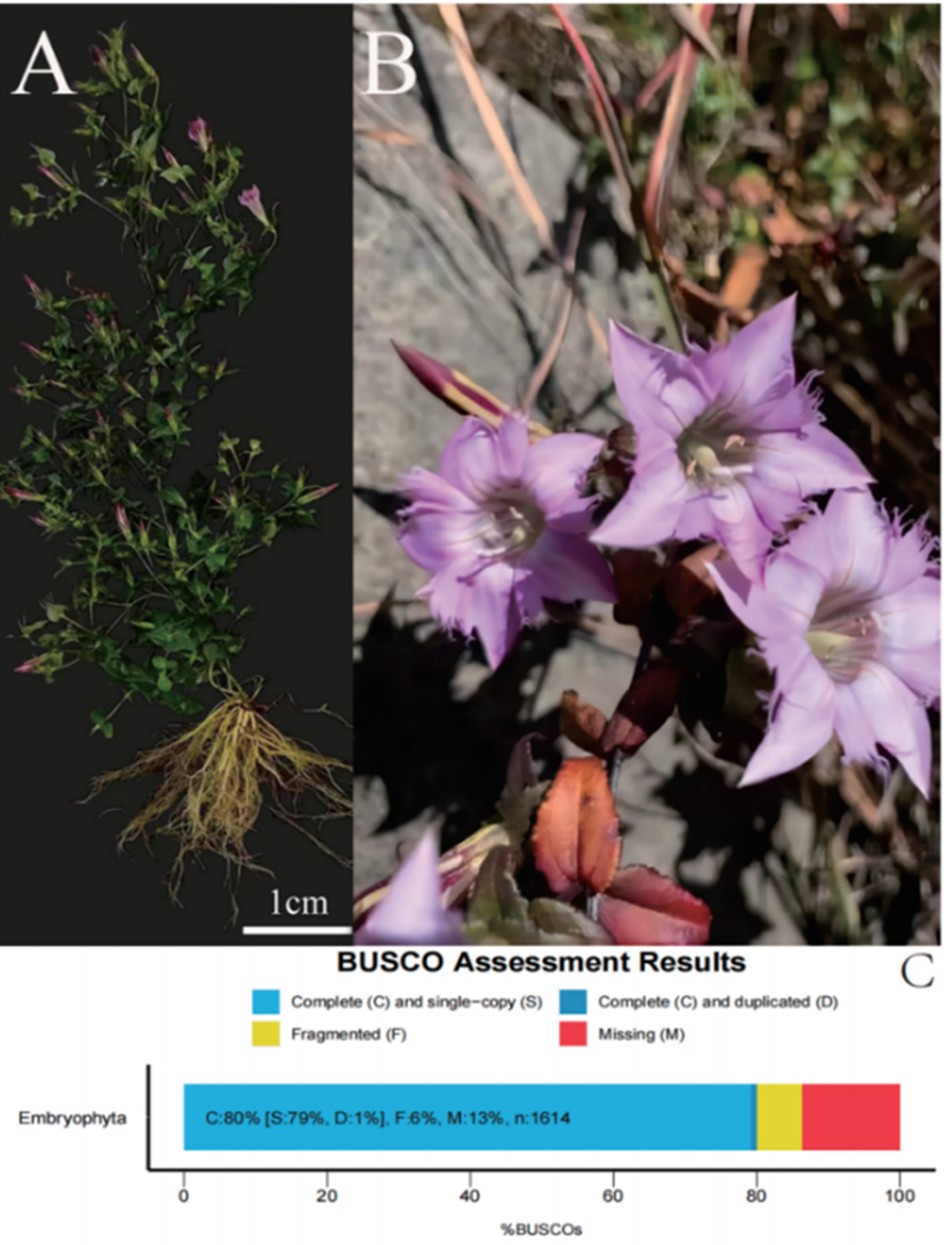

**Figure 1** **Representative images of *G.rhodantha* used for the RNA sequencing and summary of transcriptome annotations.** (A) Whole plant including root, stem, leaf, flower; wild state of *G. rhodantha*. (B) BUSCO completeness assessments of the *G. rhodantha* transcriptome. Dark blue bars represent complete and duplicated BUSCOs.

transcript to obtain the unigene (*Haas, Papanicolaou & Yassour, 2013*). Bowtie (v1.0.0)-v 0 was used to compare the sequenced reads with the unigene library. Based on the comparison results, the expression level was evaluated in combination with RSEM (v1.2.19) using the parameters (-a -m 200). The expression abundance of the corresponding unigene was expressed by the FPKM (fragments per kilobase of transcript per million mapped reads)

value. FPKM (*Trapnell et al., 2010*) is a commonly used gene expression level estimation method in transcriptome sequencing data analysis. It can eliminate the effect of differences in gene length and sequencing amount on the computational expression. The FPKM calculation method is as follows:

$$FPKM = \frac{\text{cDNA Fragments}}{\text{Mapped Fragments (Millions)} * \text{Transcript Length (kb)}}$$

Note: In the formula, cDNA fragments indicates the number of fragments as compared to a certain transcript, the number of double-ended reads; mapped fragments (millions) indicates the total number of fragments as compared to a transcript (in $10^6$ units); transcript length (kb): transcript length (in $10^3$ bases).

The transcriptome assembly was assessed in terms of their completeness and the percentage of complete, fragmented, and missing fragments by using BUSCO (v5.3.2). Parameter: -c 64 -m tran –offline -f -l embryophyta_odb10. (https://busco.ezlab.org) (*Simão et al., 2015*).

## Function annotation

The DIAMOND software (v2.0.4) (https://github.com/bbuchfink/diamond) (*Buchfink, Xie & Huson, 2015*) was used to compare the unigene sequence with the following databases: Nr (https://ftp.ncbi.nlm.nih.gov/blast/) (*Deng, Li & Wu, 2006*), Swiss-prot (http://www.uniprot.org/) (*Apweiler et al., 2004*), Clusters of Orthologous Genes (COG) (https://www.ncbi.nlm.nih.gov/COG/) (*Tatusov et al., 2000*), euKaryotic Orthologous Groups (KOG) (https://ftp.ncbi.nih.gov/pub/COG/KOG/) (*Koonin et al., 2004*), eggNOG (http://eggnogdb.embl.de/) (*Huerta-Cepas et al., 2015*) and KEGG (http://www.genome.jp/kegg/) (*Kanehisa et al., 2004*). KOBAS 2.0 (http://kobas.cbi.pku.edu.cn/) (*Xie et al., 2011*) was used to get the KEGG Origin result of the unigene in KEGG. After predicting the amino acid sequence of the unigene, the Hammer (v3.1b2) (http://hmmer.org/) (*Eddy, 1998*) software was used to compare with the Pfam (https://www.ebi.ac.uk/interpro/entry/pfam/#table) (*Finn et al., 2013*) database to obtain the annotation information of unigenes. The whole transcript data set can be found in the National Center for Biotechnology Information (NCBI) database (BioProject ID: PRJNA816320).

## Screening for secoiridoid biosynthesis genes

By referring the relevant secoiridoid biosynthesis metabolism pathways (*Wu & Liu, 2017*; *Yang, Fang & Li, 2018*), the results of nine database annotations were combined. The secoiridoid biosynthesis-related unigenes in the *G. rhodantha* transcription data were uncovered. The direct embodiment of a gene expression level is the abundance of its transcript. The higher the transcript abundance, the higher was the gene expression level. After referring the secoiridoid biosynthesis pathway in *G. scabra*, *G. rigescens*, and *G. macrophylla*, a possible biosynthetic pathway of *G. rhodantha* was speculated. Combined with the annotation results in the Nr and KEGG databases, the secoiridoid biosynthesis-related unigene in the transcriptome data was mined, and the expression amount was calculated using the TPM (transcripts per million) value.

## Identification of SSRs

The MISA (v1.0) (http://pgrc.ipk-Gatersleben.de/misa/misa.html) software was used to identify the SSR motif. The input file is a unigene sequence, and six types of SSRs were identified: (1) mono-nucleotide repeating SSR, (2) di-nucleotide repeating SSR, (3) tri-nucleotide repeating SSR, (4) tetra-nucleotide repeating SSR, (5) penta-nucleotide repeating SSR, and (6) hexa-nucleotide repeating SSR.

## Differential expression analysis

DESeq2 (v1.6.3) (*Love, Huber & Anders, 2014*) was used for differential expression analysis between samples to obtain the differential gene expression set of the two conditions. In the process of differential expression analysis, the Benjamini–Hochberg method was used to correct the significance ($p$-value) obtained by the original hypothesis test. Finally, the corrected $p$-value and False Discovery Rate (FDR) were adopted as the key indicator of differential expression gene screening. During the screening process, FDR <0.01 and FC |(fold change)| ≥2 were used as the screening criteria.

## qRT-PCR analysis

The *8-HGO* was screened from the secoiridoid biosynthesis pathway using the screening criteria of $|\log2 (FC)| \geq 2$ and FDR <0.01. The 18S rRNA is present in the ribosomal subunit, and its encoding gene rDNA (18S rRNA/rDNA) is evolutionarily conserved. The relative expression of *8-HGO* from four organs was verified with 18S rRNA as the internal reference gene. qRT-PCR was performed using the Analytik Jena-qTOWER2.2 (Analytik Jena, Jena, Germany) with TUREscript 1st Stand cDNA SYNTHESIS Kit (Aidlab, Hong Kong). Gene-specific primers were designed using Primer Premier 5.0, and the primer sequences are listed in the (Table 1). The relative gene expression was calculated by the $2^{-\Delta\Delta Ct}$ method (*Pfaffl, 2001*).

## Measuring the mangiferin, swertiamarin, and loganic acid contents

The mangiferin, swertiamarin, and loganic acid contents were estimated using the 1,260 high-performance liquid chromatography (HPLC) (Agilent, Santa Clara, CA, USA). The extraction and measurement of these three components in *G. rhodantha* were conducted as per our method for *G. rigescens* and *Liu et al. (2022a)*. First, 2.5 g of dried powder was extracted under ultrasonication in 25 ml 80% methanol for 40 min (power 150 W, working frequency 55 kHz), followed by centrifugation. Chromatographic conditions were: (1) chromatographic column: Agilent Intersil-C18 column (4.6 mm ×150 mm, 5 μm), (2) mobile phase: 0.1% formic acid aqueous solution (A) and acetonitrile (B), flow rate: one mL/ min, gradient elution: (0-—2.5 min, 7–10% B; 2.5–20 min, 10–26% B; 20–29.02 min, 26–58.3% B; 29.02–30 min, 58.3–90% B; 30–34 min, 90% B), (3) column temperature: 30 °C, (4) sample size: 5 μL, and (5) detection wavelength: 241 nm. The mangiferin, loganic acid, and swertiamarin contents in the different organs were determined by comparing their peak times and retention times with that of the standard. Finally, the data was processed using Excel 2016 and plotted using Graphpad Prism 6.01.
**Table 1  Primers of PCR.**

| Gene name | Primers sequences(5′ → 3′) | Annealing temperature (TM) |
|---|---|---|
| 18S-F | CAACCATAAACGATGCCGA | 60 ° C |
| 18S-R | AGCCTTGCGACCATACTCC | 60 ° C |
| 8-HGO-F | GAAGAAGTGAAGGACCTCAAG | 60 ° C |
| 8-HGO-R | CGGGAGCATAAATTCGTCTT | 60 ° C |

**Table 2  Summary of the transcript statistics generated from *G. rhodantha*.**

| | NO. | 300–500 bp | 500–1,000 bp | 1,000–2,000 bp | N50 Length/bp | Mean Length/bp | Total Length/bp |
|---|---|---|---|---|---|---|---|
| Transcript | 116,304 | 24,281 | 28,303 | 35,811 | 2,011 | 1,424.26 | 165,647,102 |
| Unigene | 47,871 | 18,471 | 12,482 | 9,124 | 1,826 | 1,107.38 | 53,011,260 |

# RESULTS

## Illumina sequencing and read assembly

We obtained a total of 12 libraries from the four organs, including high-quality reads fragments from 20,423,964, 21,194,626 and 21,917,414 of root, 21,475,005, 20, 386,021, and 19,271,209 of stem, 20,873,444, 21,275,398, and 20,960,228 of leaf and 21,230,658, 21,811,288, and 22,615,547 of flower. After sequence assembly, a total of 47,871 unigenes were obtained. The length of N50 was 1,826 bp, with an average length of 1,107.38 bp (Table 2). Pearson's correlation coefficient (r) was used as an indicator of studying inter-sample correlation (*Schulze, Kanwar & Gölzenleuchter, 2012*). The closer $r^2$ is to 1, the stronger was the correlation between the two samples (Fig. 2B). The figure shows that the correlation of three repeats within both the stem and root was >0.8, which indicated high reproducibility. However, the correlation of three repeats within both leaf and flower was not good. PCA (Fig. 2A) analysis showed similar results as the heat maps. A BUSCO analysis was performed to evaluate the completeness, and we recovered 1,292 of the 1,614 conserved eukaryotic genes (80%) (Fig. 1C).

## Functional annotation

After comparing and annotating the unigenes in the nine databases, (including GO, KEGG, and NR), we annotated 31,516 (65.8%) unigenes in at least one database. The specific results were GO annotation 24,320 (50.8%), KEGG annotation 19,547 (40.8%), and NR annotation 30,452 (63.6%) (Table 3).

When GO was used to classify gene functions, we assigned 24,320 unigenes to three categories: biological process, cellular component, and molecular function, with a total of 43 branches. Within the 'biological process' category, the most enriched categories were the cellular process (13,249, 54.5%), metabolic process (12,440, 51.2%), and biological regulation (3,798, 11.5%). Within the cell components, the most enriched categories

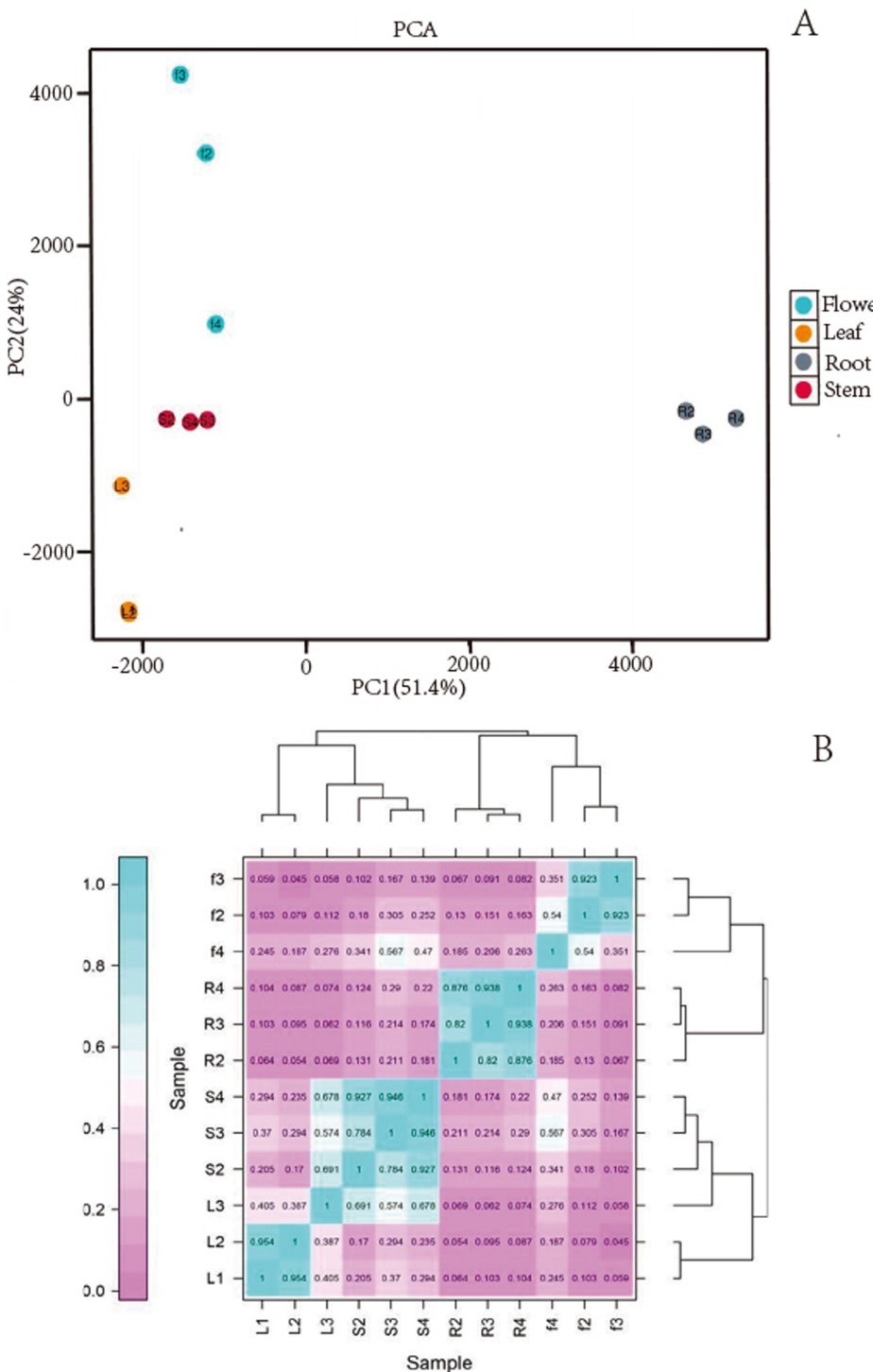

**Figure 2** (A) Images of PCA and heatmap showing the similarity and distance of samples. (B) Images of PCA heatmap; R: root; S: stem; L: leaf; F: flower.

**Table 3** Functional annotation results for *G. rhodantha*.

| Databases | Number | 300 ≤ length | Length ≥ 1,000 |
| --- | --- | --- | --- |
| COG | 9,846 | 4,572 | 5,274 |
| GO | 24,320 | 11,417 | 10,735 |
| KEGG | 19,547 | 8,812 | 9,162 |
| KOG | 16,889 | 7,727 | 9,162 |
| Pfam | 22,635 | 10,143 | 12,492 |
| Swissprot | 18,472 | 7,525 | 10,947 |
| TrEMBL | 28,198 | 13,142 | 15,056 |
| eggNOG | 23,134 | 10,017 | 13,117 |
| Nr | 30,452 | 15,261 | 15,191 |
| ALL | 31,516 | 16,144 | 15,371 |

were cellular analytical entity (13,854, 57.0%), intracellular (8,597, 35.3%), and protein-containing complex (2,568, 10.6%). Furthermore, in the molecular function category, the main branches were binding (12,033, 49.5%), catalytic activity (11,087, 45.6%), and structural molecular activity (1,544, 6.3%) (Fig. 3).

Comparing the unigenes of *G. rhodantha* with the NR database, we annotated 30,452 unigenes, which showed the highest similarity with *Coffea arabica* (4,900, 16.09%), *C. eugenioides* (2,557, 8.40%), and *C. canephora* (1,767, 5.80%) (Fig. 4).

TFs can activate the expression of multiple genes in an specific metabolic pathway, which consequently regulate the production of target metabolites (*Sun et al., 2018*). In the *G. rhodantha* transcriptome, we divided 1,005 unigenes into 66 TFs families. Previous studies showed that plants mainly have six terpenoid metabolism-related TFs families (C2H2, AP2/ERF, bHLH, MYB, NAC, and bZIP) (*Xu et al., 2019*). Here, C2H2 transcription factor family members were the most abundant (84, 8.9%), followed by AP2/ERF-ERF (65, 6.5%), bHLH (57, 5.6%), bZIP (54, 5.4%), NAC (53, 5.3%), and GRAS (51, 5.1%) (Fig. 5A).

There were significant differences in TFs in different organs. We divided the 634 root-specific unigenes into 65 TF families. AP2/ERF-ERF TF family members were the most abundant (47, 7.41%), followed by C3H (40, 6.31%), C2H2 (40, 6.31%), MYB-related (37, 5.84%), and bHLH (36, 5.68%). Furthermore, we divided the 675 stem-specific unigenes into 65 TFs families. Among them, AP2/ERF-ERF TFs family members were the most abundant (52, 7.70%), followed by C2H2 (41, 6.07%), MYB-related (39, 5.78%), C3H (38, 5.63%), and WRKY (37, 5.48%). In the leaves, we divided 602 unigenes into 64 TF families. The bHLH transcription factor family was the most abundant (41, 6.81%), followed by MYB-related (38, 6.31%), C3H (37, 6.15%), AP2/ERF-ERF (35, 5.81%), and C2H2 (34, 5.65%). Finally, we categorized 648 flower-specific unigenes into 65 TF families. Here, AP2/ERF-ERF TFs family members were the most abundant (44, 6.79%), followed by C2H2 (42, 6.48%), bHLH (41, 6.33%), C3H (39, 6.02%), and MYB-related (37, 5.25%) (Fig. 5B).

We further enriched these TF families into KEGG metabolic pathways. Two unigenes encoding the C2H2 family of TFs were enriched in tropane, piperidine, and pyridine
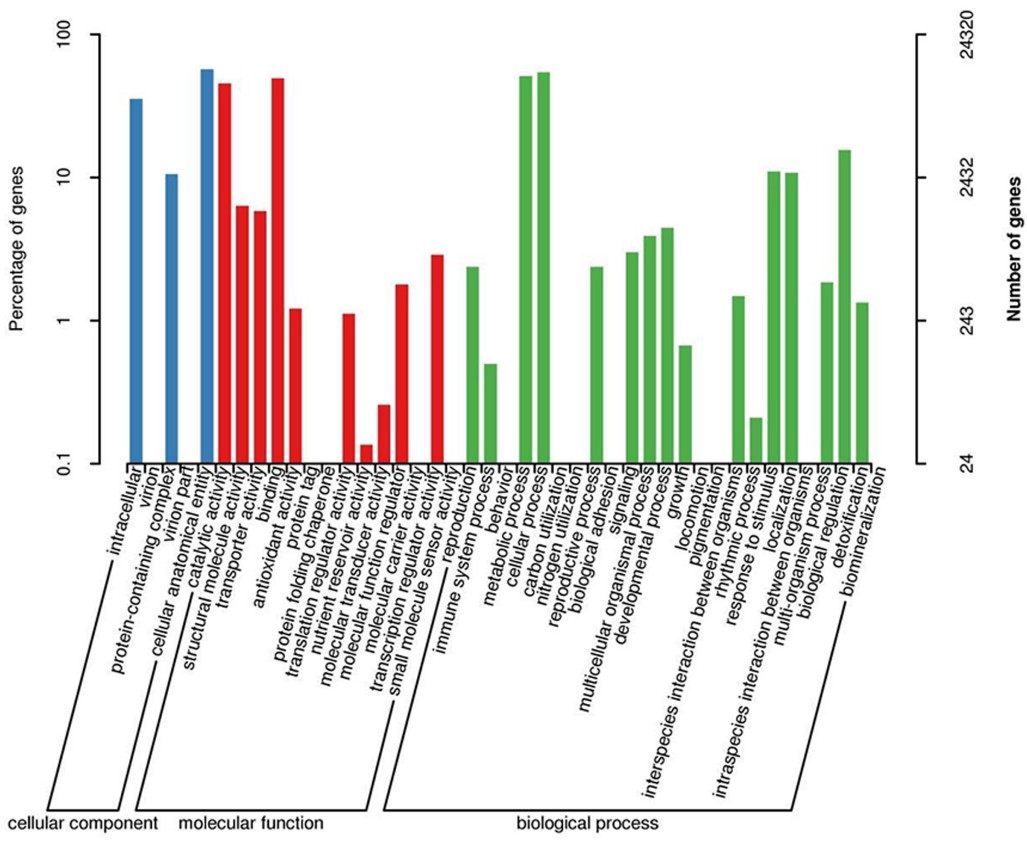

**Figure 3** GO classification map 24,320 unigenes were classified into three main categories: biological process, cellular component, and molecular function with a total of 43 branches.

alkaloid biosynthesis, whereas six unigenes encoding the zf-HD TF family members were enriched in betalain biosynthesis. Furthermore, three unigenes encoding the trihelix TF family members were enriched in ubiquinone and other terpenoid quinone biosynthesis. Finally, three unigenes encoding C3H TFs family and one encoding bHLH TFs family were enriched in the phylalanine, tyrosine, and tryptophan biosynthetic pathways (Table 4).

## Analysis of KEGG pathways

The KEGG pathways in *G. rhodantha* can be divided into five categories: cellular processes, environmental information processing, genetic information processing, metabolism, and organic systems, which we mapped to 136 KEGG pathways. The top three metabolic pathways were carbon (763, 6.95%), amino acids (510, 4.65%), and glycolysis/gluconeogenesis (432, 3.94%), with the rest being mainly enriched in pentose and gluconate interconversion along with tarch and sucrose metabolism.

The secondary metabolites contained in higher plants are closely related to their medicinal ingredients. In this regard, we assigned 1,422 unigenes to the 25 secondary metabolic pathways in *G. rhodantha*. Among them, 172 unigenes encoded key enzymes involved in the terpenoid biosynthesis pathway, including terpenoid backbone (73
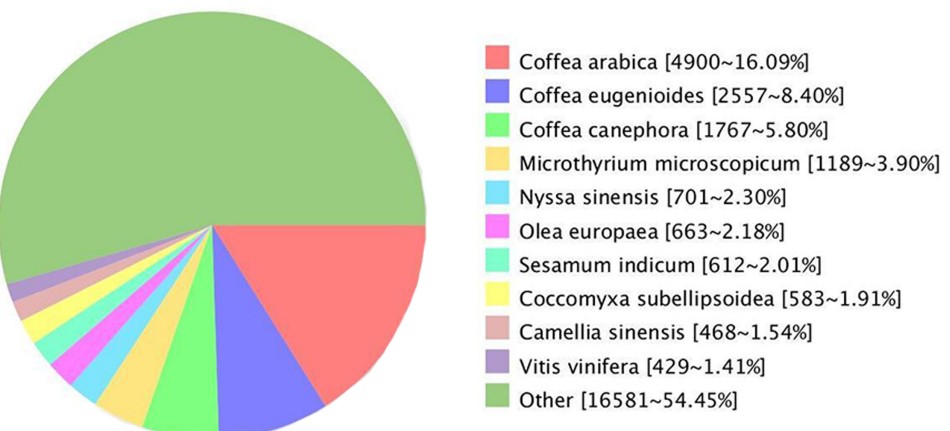

**Figure 4** **Unigenes from *G. rhodantha* distributed in Nr database.**

unigenes), monoterpenoids (21 unigenes), diterpenoids (38 unigenes), and sesquiterpenoid and triterpenoid (40 unigenes). There were 356 flavonoid biosynthesis-related unigenes, including the phenylpropanoid (253 unigenes), flavonoid (76 unigenes), flavone and flavonol biosynthesis pathways (15 unigenes), and isoflavonoid (12 unigenes). However, only 86 were alkaloid biosynthesis-associated unigenes (Table 5).

## Simple sequence repeat (SSR) analysis

SSR is one of the effective molecular markers for detecting genetic diversity and constructing a genetic map (*Liu et al., 2022a*; *Liu et al., 2022b*; *Liu et al., 2022c*). Six types of SSR were identified *via* by SSR analysis of unigenes with over 1 KB screened by MISA software: (1) mono-nucleoside repeat SSR, (2) di-nucleoside repeat SSR, (3) tri-nucleoside repeat SSR, (4) tetra-nucleoside repeat SSR, (5) penta-nucleoside repeat SSR, and (6) hexa-nucleoside repeat SSR. Thereafter, we identified a total of 7,388 unigenes, of which the mono-nucleoside and tri-nucleoside repeat SSRs received the most comments, *i.e.,* 4,573 (61.9%) and 1,385 (18.7%), respectively. They were followed by di-nucleoside, tetra-nucleoside, hexa-nucleoside, and penta nucleoside, with 982 (13.3%), 54 (7%), 24 (3%), and 14 (2%), respectively (Fig. 6A).

The identified SSRs were dominated by the A/T single-nucleotide repeats representing ~64.29%. Furthermore, the AT/TA di-nucleotide repeat type accounted for 9.45% of the total SSR. The tri-nucleotides repeat types, *i.e.,* GGT/GAT and GAA/ATA accounted for 1.62% and 1.47%, respectively. However, the penta-nucleotide and hexa-nucleotide repeats had the lowest proportion of 0.02% (Fig. 6B).

## Differential gene expression analysis

Since gene expression is spatiotemporal specific, we pairwise compared the transcriptome data from four different organs. In the process of DEGs analysis, we used the Benjamin-Hochberg method to correct the significant row *p*- value obtained from the original hypothesis test. In the screening process, FDR <0.01 and FC (fold change) $\geq$ 2 were used

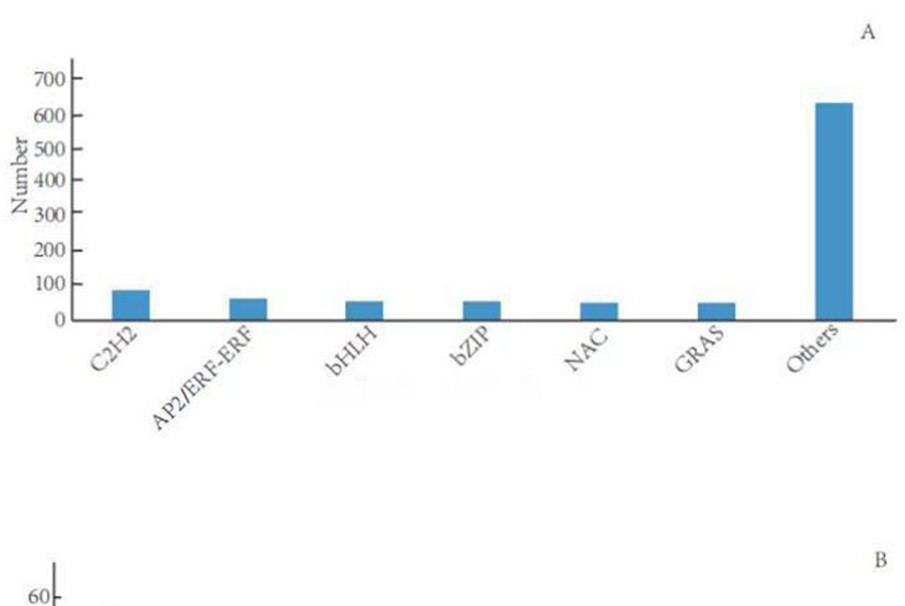

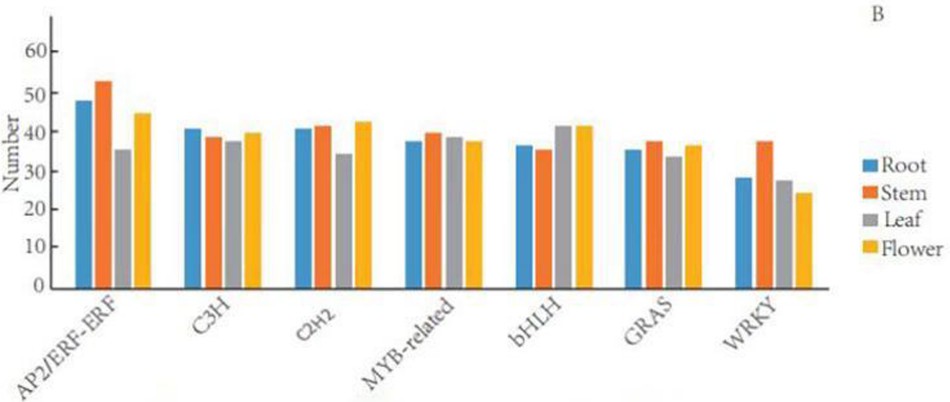

**Figure 5** **Distribution of transcription factor families in general and in different organs in *G. rhodantha*.** (A) Over distribution of TFs; (B) Distribution of TFs in different organs.

as the screening criteria. *G. rhodantha* is a perennial herbaceous grass used as medicines. Considering their roots are perennial and its aerial parts (stem, leaf, and flower) are annual, we chose root as the control group. When we compared the control group (root) with the experimental group (stem, leaf, and flower), we found significant transcript differences. The number of differential expressions was the highest between the root and flower, whereas it was the lowest between the root and leaves (Fig. 7A). A total of 4,606 transcripts showed organ-specific expression, of which 966, 2,191, 274, and 1,175 transcripts were from the root, stem, leaf, and flower, respectively (Fig. 7B).

The annotation and analysis of metabolic pathways of differentially expressed genes (DEGs) is helpful for further understanding the function of genes. When regarding root as the control group and the stem as the experimental group, 6,156 DEGs were enriched in 129 metabolic pathways, including benzoxazinoid biosynthesis (13), flavoid biosynthesis (39), stilbenoid, dialylheptanoid and ginger biosynthesis (19), circadian rhythm-plant (41), and photosynthesis-antenna proteins (24). When regarding root as the control group and leaf as

**Table 4  Transcription factors (TFs) involved in the secondary metabolites.**

| Gene ID | Pathway name | TF family |
|---|---|---|
| c160641.graph_c0 | Tropane, piperidine and pyridine alkaloid biosynthesis | C2H2 |
| c200088.graph_c0 | Tropane, piperidine and pyridine alkaloid biosynthesis | C2H2 |
| c186566.graph_c0 | Betalain biosynthesis | zf-HD |
| c190764.graph_c0 | Betalain biosynthesis | zf-HD |
| c193607.graph_c0 | Betalain biosynthesis | zf-HD |
| c195440.graph_c0 | Betalain biosynthesis | zf-HD |
| c197274.graph_c1 | Betalain biosynthesis | zf-HD |
| c203081.graph_c0 | Betalain biosynthesis | zf-HD |
| c191415.graph_c0 | Ubiquinone and other terpenoid-quinone biosynthesis | Trihelix |
| c195369.graph_c0 | Ubiquinone and other terpenoid-quinone biosynthesis | Trihelix |
| c197313.graph_c0 | Ubiquinone and other terpenoid-quinone biosynthesis | Trihelix |
| c194155.graph_c0 | Phenylalanine, tyrosine and tryptophan biosynthesis | C3H |
| c194543.graph_c0 | Phenylalanine, tyrosine and tryptophan biosynthesis | C3H |
| c197534.graph_c1 | Phenylalanine, tyrosine and tryptophan biosynthesis | C3H |
| c195973.graph_c1 | Phenylalanine, tyrosine and tryptophan biosynthesis | bHLH |

the experimental group, 4,534 DEGs were enriched in 129 metabolic pathways, which were significantly enriched in flavone and flavonol biosynthesis (8), porphyrin and chlorophyll metabolism (32), glycosphingolipid biosynthesis-ganglio series (15), glycosaminoglycan degradation (17), and flavor biosynthesis (22). Finally, when regarding root as the control group and the flower as the experimental group, 6,649 DEGs were enriched in 131 metabolic pathways, which were significantly enriched in cutin, suberine, and wax biosynthesis (22), zeatin biosynthesis (20), cyanoamino acid metabolism (33), flavor biosynthesis (27), and plant hormone signal transduction (165) (Table 6). These DEGs were significantly enriched in flavonoid biosynthesis, which showed that the flavonoid accumulation was more abundant.

## Gene expression analysis of unigenes associated with mangiferin, swertiamarin, and loganic acid biosynthetic pathway

Although mangiferin is one of the main active components of *G. rhodantha*, its biosynthesis has not been included in the KEGG database to date. Therefore, we focused on analysis and identification of the putative genes in the secoiridoid biosynthesis pathway.

The secoiridoid biosynthesis is probably completed using the following steps: intermediate generation, terpene skeleton synthesis, and post-modification (*Wu & Liu, 2017*; *Kang et al., 2021*). The analysis results showed that 54 unigenes in the *G. rhodantha* transcriptome encoding 17 key enzymes in different organs (Table 7) and related genes, which is represented by the heat map.

A wide variety of terpenoids with diverse structures are synthesized from common precursors isopentenyl diphosphate (IPP) and dimethylallyl diphosphate (DMAPP). These compounds can be derived from both the mevalonic acid (MVA) pathway in the cytoplasm and the 2-C-methyl-D-erythritol-4-phosphate (MEP) pathway in plastids (*Singh, Gahlan & Kumar, 2012*). IPP and DMAPP are catalyzed by geranyl pyrophosphate synthase

**Table 5  Secondary metabolism KEGG pathway analysis of transcriptomic unigenes in *G. rhodantha*.**

| No. | KEGG Pathway | Pathway ID | No. of unigene |
|---|---|---|---|
| 1 | Steroid biosynthesis | ko00100 | 70 |
| 2 | Ubiquinone and other terpenoid-quinone biosynthesis | ko00130 | 83 |
| 3 | Purine metabolism | ko00230 | 172 |
| 4 | Caffeine metabolism | ko00232 | 7 |
| 5 | Phenylalanine, tyrosine and tryptophan biosynthesis | ko00400 | 81 |
| 6 | Nicotinate and nicotinamide metabolism | ko00760 | 46 |
| 7 | Porphyrin and chlorophyll metabolism | ko00860 | 79 |
| 8 | Terpenoid backbone biosynthesis | ko00900 | 73 |
| 9 | Indole alkaloid biosynthesis | ko00901 | 11 |
| 10 | Monoterpenoid biosynthesis | ko00902 | 21 |
| 11 | Limonene and pinene degradation | ko00903 | 43 |
| 12 | Diterpenoid biosynthesis | ko00904 | 38 |
| 13 | Brassinosteroid biosynthesis | ko00905 | 18 |
| 14 | Carotenoid biosynthesis | ko00906 | 82 |
| 15 | Zeatin biosynthesis | ko00908 | 46 |
| 16 | Sesquiterpenoid and triterpenoid biosynthesis | ko00909 | 40 |
| 17 | Phenylpropanoid biosynthesis | ko00940 | 253 |
| 18 | Flavonoid biosynthesis | ko00941 | 76 |
| 19 | Anthocyanin biosynthesis | ko00942 | 4 |
| 20 | Isoflavonoid biosynthesis | ko00943 | 12 |
| 21 | Flavone and flavonol biosynthesis | ko00944 | 15 |
| 22 | Stilbenoid, diarylheptanoid and gingerol biosynthesis | ko00945 | 38 |
| 23 | Isoquinoline alkaloid biosynthesis | ko00950 | 43 |
| 24 | Tropane, piperidine and pyridine alkaloid biosynthesis | ko00960 | 53 |
| 25 | Betalain biosynthesis | ko00965 | 18 |

(GPPS) to geranyl diphosphate (GPP), which is an important cut-off point. The GPP flow through different metabolic directions to monoterpenes, diterpenes, triterpenes, *etc.* For the synthesis of secoiridoid, geraniol is the starting compound. There may be three pathways from GPP to geraniol (Fig. 8A). The pathway involving the transformation of GPP into geraniol and pyrophosphate by the action of geraniol synthase (GES) was fully annotated, while the other two pathways were not fully annotated. Geraniol was converted into iridodial, which was the skeleton of secoiridoids under the excessive step reaction (*Miettinen et al., 2014*; *Lichman et al., 2019*). Iridodial is converted into secoiridoid glycoside compounds under a series of modification processes involving the addition of sugar groups, deoxygenation, ring opening, *etc* (*Liu et al., 2017*; *Rain & Takahashi, 2016*).

Based on the statistics of TPM value, we searched for the TPM values of the genes in different organs and then used TBtools(v1.098) to plot the heatmap. For MVA pathway, the *AACT2* expression is relatively high in the root, stem, and flower, whereas that of *HMGS1* is relatively high in root and flower (Fig. 8B). For the MEP pathway, the relative expression of *DXR* was higher in the leaf and flower, while that of *HDR* was higher in the

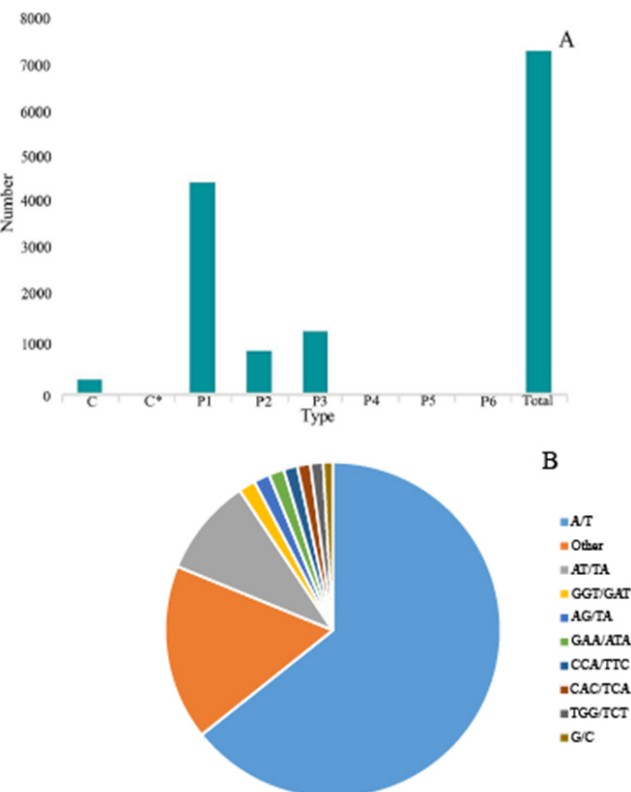

**Figure 6** **Simple sequence repeats (SSRs) in *G. rhodantha*.** (A) Distribution of different types of SSR. (B) Frequency of most abundant SSR motifs. C: Composite repetitive SSR, C*: There are overlapping composite type SSRs, P1: mono-nucleoside repeat SSR, P2: di-nucleoside repeat SSR, P3: tri-nucleoside repeat SSR, P4: tetra-nucleoside repeat SSR, P5: Penta-nucleoside repeat SSR, P6: Hexa-nucleoside repeat SSR.

aerial parts (Fig. 8C). In secoiridoid pathway, the relative expression of *8-HGO* was also higher in the aerial part (Fig. 8D).

In the gentiopicroside biosynthesis pathway, *8-HGO* is particularly important as its structural gene (*Wang, 2020*) . *8-HGO* is the key enzyme behind the secoiridoid skeleton construction. Additionally, we verified the relative expression of *8-HGO* from the four organs using qRT-PCR. These results showed a similar expression pattern with that of the transcriptome. The relative expression of the *8-HGO* gene was higher in the aerial parts than in the root (Fig. 9). Therefore, this suggested that the secoiridoid component was mainly synthesized in aerial part, especially in the leaves.

## Contents of mangiferin, swertiamarin, and loganic acid

To examine the possible relationship between the gene expression and their corresponding metabolites, we determined the content of three bioactive compounds including two secoiridoid pathway metabolites (swertiamarin and loganic acid and mangiferin. After comparing their contents, all three compounds showed similar accumulation patterns in the different organs, with the lowest levels being in the root. Furthermore, swertiamarin

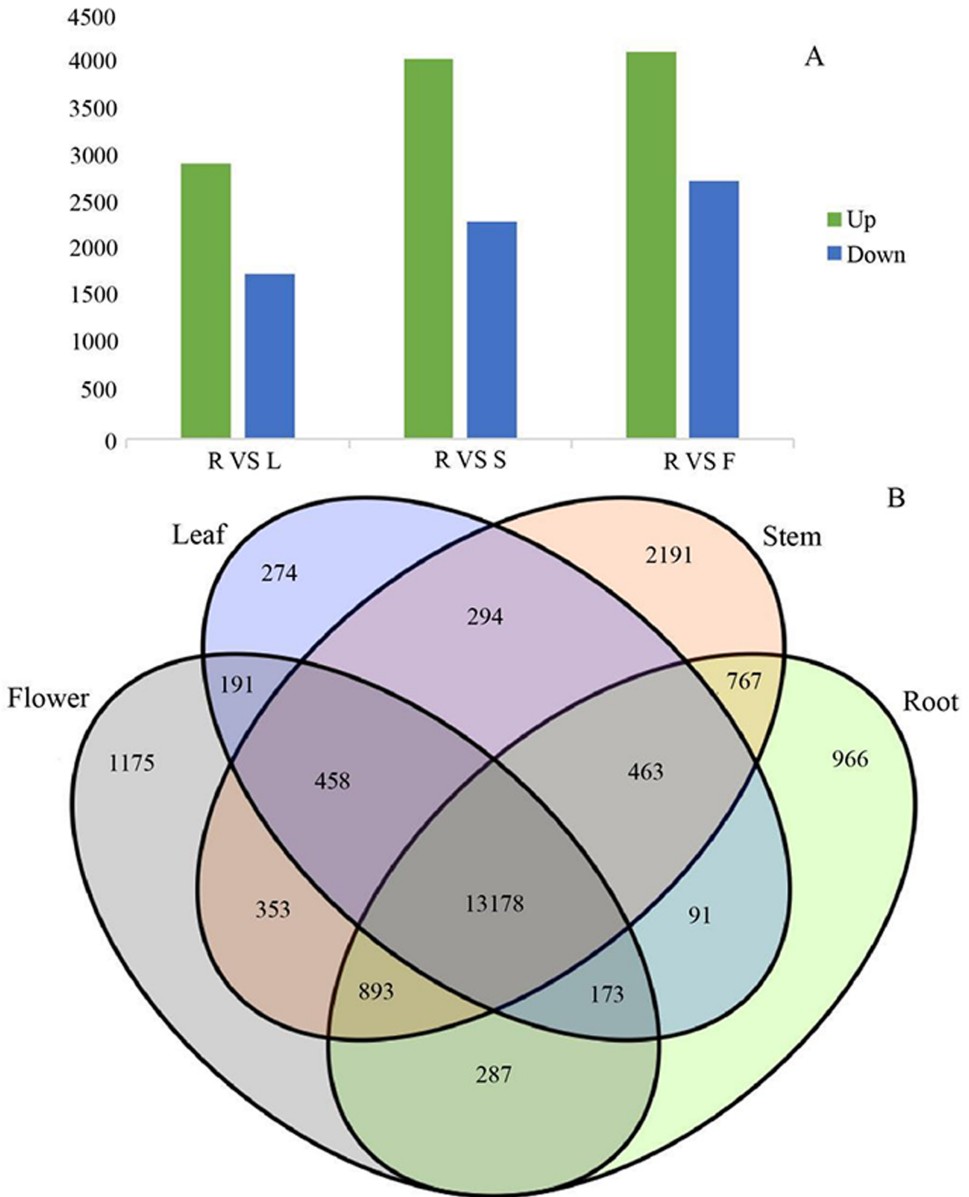

**Figure 7  Differential expression analysis of four organs in *G. rhodantha*.** (A) Number of genes of up-regulated and down-regulate while root compared with other three organs respectively. (B) Venn diagram representing the number of DEGs among all organs. R: root; S: stem; L: leaf; F: flower.

was the least abundant in the root and the most abundant in flower (Fig. 10). Therefore, we found that the contents of the three components in the aerial parts were significantly higher than in the root.

## DISCUSSION

Secoiridoid is one of the important medicinal components for *G. rhodantha* (*Li et al., 2006*; *Inao et al., 2004*). The analysis of the secoiridoid biosynthesis pathway is very important for
**Table 6  Enrichment KEGG analysis of DEGs.**

| Sample | Pathway | KEGG pathway number | Number |
|---|---|---|---|
| Root *vs* Stem | Benzoxazinoid biosynthesis | Ko00402 | 13 |
| | Flavonoid biosynthesis | Ko00941 | 39 |
| | Stilbenoid, diarylheptanoid and gingerol biosynthesis | Ko00945 | 19 |
| | Circadian rhythm - plant | Ko04712 | 41 |
| | Photosynthesis - antenna proteins | ko00196 | 24 |
| Root *vs* Leaf | Flavone and flavonol biosynthesis | Ko00944 | 8 |
| | Porphyrin and chlorophyll metabolism | ko00860 | 32 |
| | Glycosphingolipid biosynthesis - ganglio series | ko00604 | 15 |
| | Glycosaminoglycan degradation | ko00531 | 17 |
| | Flavor biosynthesis | ko00941 | 22 |
| Root *vs* Flower | Cutin, suberine and wax biosynthesis | ko00073 | 22 |
| | Zeatin biosynthesis | ko00908 | 20 |
| | Cyanoamino acid metabolism | ko00460 | 33 |
| | Flavone biosynthesis | ko00941 | 27 |
| | Plant hormone signal transduction | ko04075 | 165 |

**Table 7  Unigenes involved in biosynthesis of secoiridoid.**

| Pathway | Enzyme | Abbr. | EC | No. of unigene |
|---|---|---|---|---|
| MVA | Acetyl-CoA *C*-acetyltransferase | AACT | 2.2.1.9 | 9 |
| | Hydroxymethylglutaryl-CoA synthase | HMGS | 2.3.3.10 | 2 |
| | Hydroxymethylglutaryl-CoA reductase (NADPH) | HMGR | 1.1.1.34 | 4 |
| | Isopentenyl phosphate kinase | IPK | 2.7.4.26 | 1 |
| | Mevalonate kinase | MK | 2.7.1.36 | 1 |
| | Phosphomevalonate kinase | PMK | 2.7.4.2 | 3 |
| | Diphosphomevalonate decarboxylase | MVD | 4.1.1.33 | 1 |
| MEP | 1-Deoxy-*D*-xylulose-5-phosphate synthase | DXS | 2.2.1.7 | 3 |
| | 1-Deoxy-*D*-xylulose-5-phosphate reductoisomerase | DXR | 1.1.1.267 | 1 |
| | 2-*C*-Methyl-*D*-erythritol 4-phosphate cytidylyltransferase | CMS | 2.7.7.60 | 1 |
| | 4-(Cytidine 5′-diphospho)-2-*C*-methyl-*D*-erythritol kinase | CMK | 2.7.1.148 | 1 |
| | 2-*C*-Methyl-*D*-erythritol 2,4-cyclodiphosphate synthase | MCS | 4.6.1.12 | 1 |
| | 4-Hydroxy-3-methylbut-2-enyl-diphosphate synthase | HDS | 1.17.7.3 | 1 |
| | 4-Hydroxy-3-methylbut-2-enyl diphosphate reductase | HDR | 1.17.7.4 | 1 |
| secoiridoid | 8-Hydroxygeraniol oxidoreductase | 8-HGO | 1.1.1.324 | 6 |
| | Strictosidine synthase | STR | 4.3.3.2 | 8 |
| | Geranylgeranyl pyrophosphate synthase | GGPPS | 2.5.1.29 | 10 |

quality improvement and breeding for *G. rhodantha*. First, we sequenced the transcriptome of different organs of *G. rhodantha*. When comparing with other plants of the same genus, the results showed that sequence splicing quality of *G. rhodantha* was relatively high (47,871 unigenes obtained, average length 1,107.38 bp), as compared with *G. rigescens*

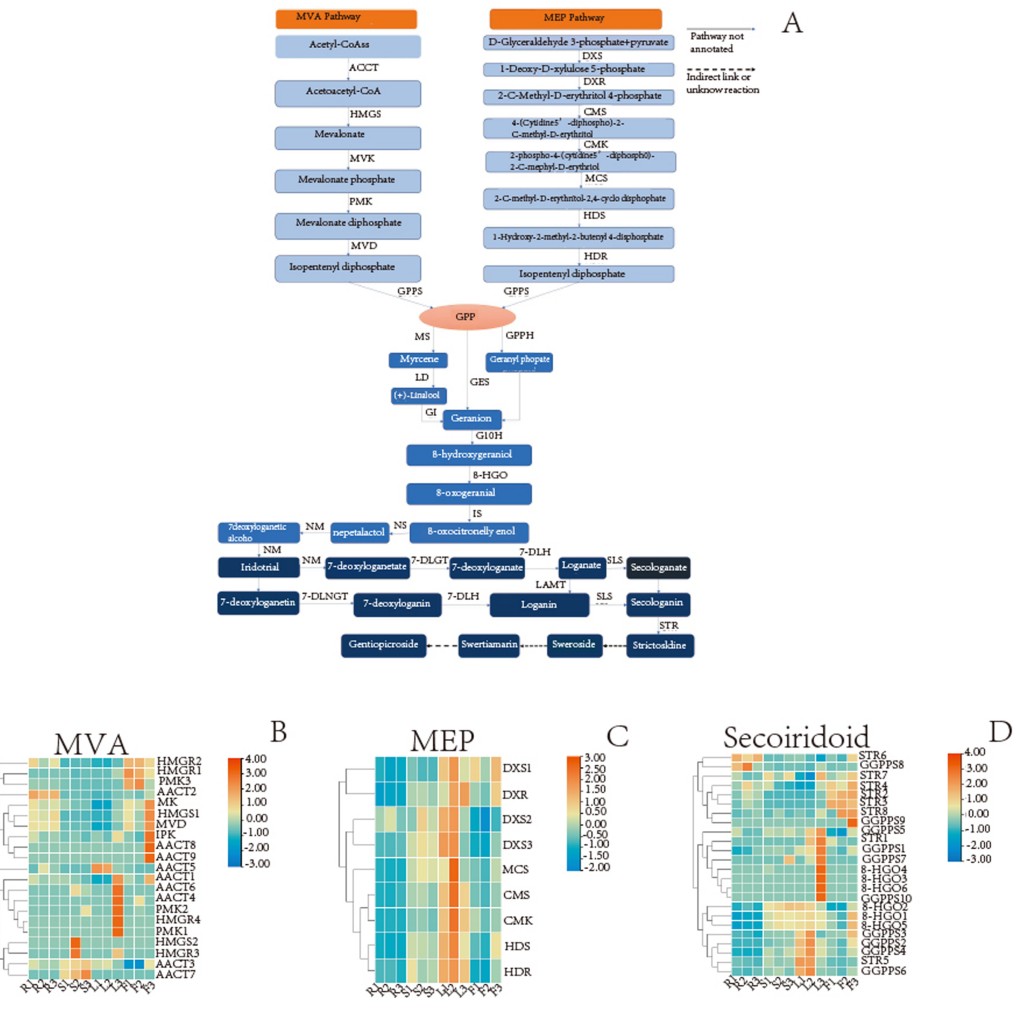

**Figure 8** Detailed pathways and candidate unigenes involved in the mevalonate (MVA) pathway or methylerythritol phosphate (MEP) pathway route of secoirdoid biosynthesis and their expression levels. (A) Proposed pathway of secoiridoids biosynthesis. (B) the DEGs in the MVA pathway (C) the DEGs in the MVA pathway (D) the DEGs from GPPs to secoiridoid.

(76,717 unigenes obtained, average length 753 bp) (*Zhang et al., 2015*), *G. crassicaulis* (159,534 unigenes obtained, average length 679 bp) (*Kang et al., 2021*), *G. waltonii* (79,455 unigenes obtained, average length 834 bp) (*Ni et al., 2019*).

There were 30,452 unigenes annotated in the Nr database for *G. rhodantha*, with the results showing the highest similarity to *C. arabica* from the Rubiaceae family. This may be due to the limited data about Gentianaceae. Based on the results of the DEGs of *G. rhodantha*, we found more metabolic pathways in the root with respect to flower than in the other contrast pairs (root with respect to stem, root with respect to leaf). The DEGs were mainly enriched in phytohormone signaling, flavonoid biosynthetic pathways, and cyanogenic amino acid metabolism. Higher plants contain diverse secondary metabolites, which are closely related to their medicinal effects. Additionally, mangiferin, swertiamarin,

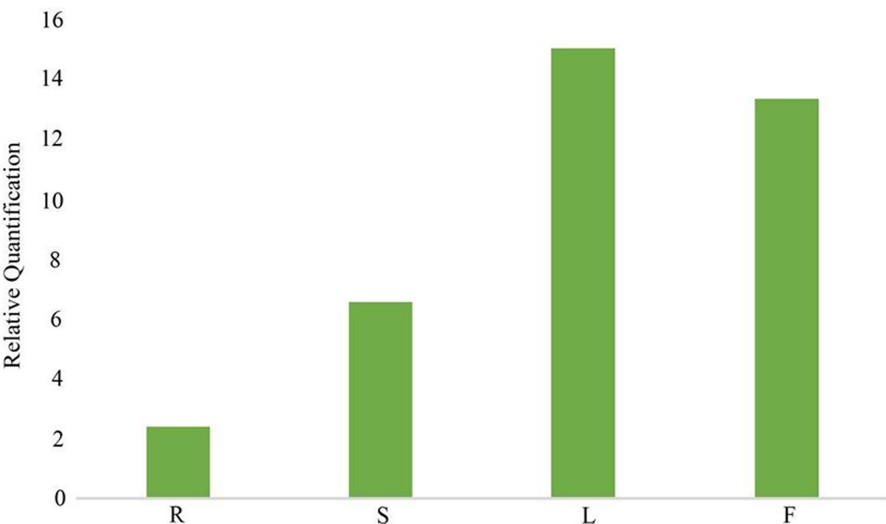

**Figure 9** Relative expression of *8-HGO* in different organs. R: root; S: stem; L: leaf; F: flower.

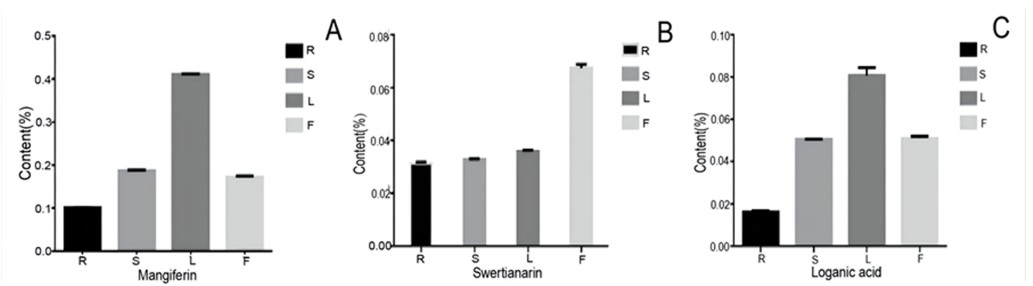

**Figure 10** The content of mangiferin, swertiamarin, and loganic acid in diûerent tissues. R, root; S, stem; L, leaf; F, flower.

and loganic acid showed similar accumulation patterns in the different organs, with the lowest levels seen in the root. This may be because the accumulation of the extremely bitter tasting swertiamarin in the aerial parts, especially in the flower, could effectively defend against predator aggression. Mangiferin showed the highest accumulation in all the organs, especially in the leaf, which was consistent with the previous studies (*Zhang et al., 2007*).

The synthesis of secondary metabolites is a complex multi-step process (*Wu et al., 2022*). KEGG analysis showed that 1,422 unigenes were involved in 25 secondary metabolic pathways, including isoquinoline alkaloid biosynthesis (ko00905), dieterpenoid biosynthesis (ko00904), and sesquiterpenoid and triterpenoid biosynthesis (ko00909). In this study, we selected *8-HGO* from the secoiridoid biosynthesis pathway for qRT-PCR validation experiments, and the results were consistent with the expected results. The low content of these three components in the root showed a correlation with the relatively low expression of *8-HGO*, thereby confirming the presence of *8-HGO* in the secoiridoid biosynthetic pathway. This was consistent with previous reports about *G. rhodantha* and
those of other species of Gentianaceae, like *Swertia mussotii* (*Shen et al., 2016*; *Liu et al., 2017*). Based on above research, it can be inferred that this active composition is mainly synthesized in the aerial part, with the medicinal value of the aerial part being higher than the root. Moveover, the biomass of the roots is very small. When used as a medicinal material, we recommend that the medicinal part should be changed to the aerial part for better protection of the resources and the maintenance of sustainable use.

TFs can control gene expression by specifically binding with the cis-regulatory elements in the promoter region of target genes, and play a key regulatory role in the plant growth and development (*Latchman, 1997*). Currently, hundreds of TF families have been isolated and identified from higher plants, which are closely related to plant stress resistance, and can regulate the expression of genes related to different plant stressors, like drought, high salt, low temperature, and pathogens (*Gibbs et al., 2015*; *Verma, Ravindran & Kumar, 2016*). The number of genes encoding different TFs families varies in different plant species, and they often have species-tissue-specific or developmental stage-specific function(s) (*Singh, 1998*). A total of 1,005 unigenes were involved in encoding 66 TF families for *G. rhodantha*. The members of the TF families did not exhibit any significant organ-specificity. Among all the TFs, the C2H2 transcription factor family was the most abundant (89, 8.9%). C2H2 zinc finger proteins play roles in the plant response to a diverse stresses, including low temperatures, salt, drought, oxidative stress, excessive light, and silique shattering (*Kim et al., 2013*; *Yue et al., 2016*; *Jiang et al., 2022*). C2H2 may also be important in the synthesis of secondary metabolites involved in the stress resistance of *G. rhodantha*. It can be considered for the study of transcription factors regulating the quality of *G. rhodantha*. Furthermore, 141 unigenes were annotated as WRKY family transcription factors, of which 17 showed higher expressions in the leaf than in the root. WRKY may be a good candidate for studying the secoiridoid biosynthesis regulation in *G. rigescens* (*Zhang et al., 2015*). The identification of these transcription factors will be helpful to further analyze the molecular mechanism of secoiridoid biosynthesis and lay a foundation for regulating the secoiridoid metabolite accumulation. Therefore, these findings are highly significant as they provide a reference for mining the key genes of the biosynthesis pathway of secondary metabolites.

As far as the reported transcriptome analysis of *G. rhodantha* is concerned, it is only the initial stage, and more extensive research is necessary. Furthermore, the expression of genes related to secondary metabolites is also affected by many factors, including plant growth and developmental stage and the ecological environment. Since we had only studied the flowering period, therefore, we could not capture all the gene expression-related information.

Simple repeats, also known as short tandem repeats, are 1–6 nt long DNA sequences widely distributed in the eukaryotic genomes. Six nucleotides form repetitive motifs in different orders. Since the motifs are repeated several times, so they have repeatability, polymorphism, richness, and co-dominance (*Chen et al., 2012*; *Shi et al., 2016*). We found a total of 7388 SSR loci in the transcriptome of *G. rhodantha*, including various nucleotide types, thereby indicating that the SSR loci of *G. rhodantha* was rich and abundant. The number of single-nucleotides repeats of A/T is the largest (4,573), which was consistent with the previous results of *Gardenia jasminoide* (*Liu et al., 2022a*; *Liu et al., 2022b*; *Liu et*

*al., 2022c*). Unfortunately, the SSR data for other species of the *Gentiana* genus have not been reported yet. Therefore, our study presents information for the further development of SSR molecular markers and the DNA ID code construction of *G. rhodantha*.

## CONCLUSION

In this study, we obtained the transcriptome data of *G. rhodantha* by high-throughput sequencing for the first time, and then determined the genes and DEGs involved in the secoiridoid biosynthesis pathway. Using qRT-PCR, we further verified the RNA-seq analysis results for one key enzyme gene related to secoiridiod biosynthesis. Therefore, our findings provide timely clues for a better understanding of the molecular mechanism of secoiridoid biosynthesis in *G. rhodantha*.

### Funding

This work was supported by the National Natural Science Foundation of China (30260110037) and the Joint special program of traditional Chinese medicine in Yunnan Province–General Program (30272110111). The funders had no role in study design, data collection and analysis, decision to publish, or preparation of the manuscript.

### Grant Disclosures

The following grant information was disclosed by the authors:
National Natural Science Foundation of China: 30260110037.
Yunnan Province–General Program: 30272110111.

### Competing Interests

The authors declare there are no competing interests.

### Author Contributions

- Ting Zhang conceived and designed the experiments, performed the experiments, analyzed the data, prepared figures and/or tables, authored or reviewed drafts of the article, and approved the final draft.
- Miaomiao Wang performed the experiments, prepared figures and/or tables, and approved the final draft.
- Zhaoju Li performed the experiments, prepared figures and/or tables, and approved the final draft.
- Xien Wu performed the experiments, analyzed the data, prepared figures and/or tables, and approved the final draft.
- Xiaoli Liu conceived and designed the experiments, performed the experiments, analyzed the data, prepared figures and/or tables, authored or reviewed drafts of the article, and approved the final draft.

## Data Availability

The sequences are available at NCBI: PRJNA816320.

## Supplemental Information

Supplemental information for this article can be found online at http://dx.doi.org/10.7717/peerj.14968#supplemental-information.

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
