# Peer review of "Transcriptome analysis and exploration of genes involved in the biosynthesis of secoiridoids in Gentiana rhodantha"

_PeerJ, doi:10.7717/peerj.14968_

## Round 0.1 · original submission · Major Revisions

The manuscript contains good information but needs exhaustive revision.

·

Basic reporting

This is a significant and new development with respect to this medicinal plant, Gentiana rhodantha. The authors have generated and analyzed transcriptome data for Gentiana rhodantha, which in itself is a significant technical advance, as Gentiana rhodantha is a less studied species with extremely scanty genomic information. The present paper shall serve as an index paper to the RNAseq data for this species. I have checked that the authors have deposited the data in a public repository for benefit of the scientific community.
Apart from the raw data, authors have analyzed the data, with standard bioinformatics tools and presented important findings with respect to the gene function of this rare species, which is commendable.
Overall, I recommend the manuscript for publication, after minor revision.

Experimental design

The experimental design is standard and acceptable in the field. Authors may elaborate in detail about de novo assembly part as reference genomic information is not available for Gentiana rhodantha, unlike other species. Therefore, greater description about de novo analysis part may be elaborated.

Validity of the findings

The authors say that they have sequenced the transcriptome of different tissues of G. rhodantha, and compared it with with Gentiana crassicaulis Duthie ex Burk.; Gentiana waltonii Burk.
Is the functional analysis reported here is de novo or with these two species as reference, is not clear.
The authors should elaborate about genomic information in the other species of Gentiana in the introduction. The authors should contextualize it with de novo analysis, which they claim.
Also, phamacological descriptions of other Gentiana species should be given in introduction.

Additional comments

In the introduction part, literature about prior genomics work on Gentiana rhodantha should be mentioned and cited. I checked that in BioProject, there are 3 other related entries with respect to Gentiana rhodantha. So, prior genomic work should be cited.

I found minor, but many copy editing type mistakes in the text. Therefore, authors should check the text carefully for proper formatting, prior to re-submission

The English language is acceptable, given the non-native speaking authors.

All the figures, in the pdf available for review, are a little blurred and could not be accepted in the final publication. Authors and production manager to ensure that resolution of figures is acceptable in the final version.

Reviewer 2 ·

Basic reporting

The manuscript of Zhang et al. presents a transcriptional study of Gentiana rhodantha. The study is novel, as it is the first transcriptome-wide study for the species (for which there is also no reference genome). However, the manuscript as it is seems more like a preliminary version than a finished version of a scientific paper. I recommend a careful revision before re-submitting it. The paper also needs a language review by an English speaker.

I attach my comments, detailing first the major ones, then the minor ones, to contribute to the future publication of the paper:
- Indicate relevant characteristics of the Gentiana rhodantha genotype studied. Why was it selected? Why is that genotype interesting and not another? Is it a genebank accession, an inbred material, a commercial genotype?
- MyM indicates: "Fresh roots, stems, leaves and flowers with three replicates of each tissue, quickly frozen in liquid nitrogen". But there is no indication of the trial or experimental study carried out to generate that sample. How did the plants grow? At what age were they sampled? I imagine, from the way a good RNAseq design is done, that the replicates of each tissue correspond to different plants. However, the way the text is laid out, it looks like everything came from a single plant. Please review and, if appropriate, correct.
- For the de novo assembly, the authors specify to have used Trinity, with default parameters. There are extensive reports on the difficulty of making de novo assembly of plant transcriptomes reliable. What other parameters/software did you try to compare, and determine that this is optimal? What post-filters did you apply to select reliable transcripts? What filters were used to eliminate possible contaminants?
- While this work is the first transcriptome for G. rhodantha, it is not the first for Gentiana. In 2015, the transcriptome of G rigescens was published (https://doi.org/10.3390%2Fijms160511550). The authors omitted this important background for the species. It would be interesting to make the comparison of both transcriptomes, as an approach to validate the results obtained here. Another strategy is to compare the unigenes assembled here with those available in NCBI for the species, and thus corroborate if, for example, the lengths of the assembled genes coincide with what is expected.
- In the section "differential expression analysis" the authors state: "Genes with an adjusted P-value <0.05 found by DESeq were assigned as differentially expressed". But in Results they say "FDR < 0.01 and fold change > 2 were used as the screening criteria". What was the criterion applied?
- The authors specify that as an internal qPCR control they used 18S. One of the big issues in qPCR is to obtain good internal controls, for which multiple studies are being done. Is there already a report on the use of 18S in the species? If so, please specify the citation. If not, explain why you selected it, and what you have done to verify that it is really a suitable reference gene for gene expression normalization in this species.
- On lines 176-177, the authors state: "when we compare the control group (the root) with the experimental group (the stem, leaf, flower) we can find significant transcriptional differences". This is the first time in the article that it is mentioned that root is the control group. Why is root the control? What is the purpose of this comparison? It is not clear from the article
- The authors should revise and redo all the figures. The graphs do not meet the requirements to be self-explanatory.

Minor comments
- On line 76, the authors specify "After the transcriptome was assembled, clean reads were compared with the transcriptome". Delete that line from that section.
- On line 81, the authors specify "Raw data (raw reads) of fastq format were firstly processed through in-house perl scripts." For the article to be FAIR, the authors must make those scripts public. I recommend sharing them in a Github-like repository or, failing that, as a supplementary file.
- Expand the section "Identification of SSRs" with the parameters established in MISA to determine SSRs. Also, expand the results and discussion of this topic. I recommend further reading on Molecular markers.

Experimental design

All of my comments were previously detailed

Validity of the findings

All of my comments were previously detailed

Additional comments

All of my comments were previously detailed

Reviewer 3 ·

Basic reporting

In this manuscript titled “Transcriptome analysis and exploration of genes involved in the biosynthesis of secoiridoids in Gentiana rhodantha” by Zhang et al., the authors constructed the transcriptome of the herb Gentiana rhodantha and attempted to explore candidate genes involved in the biosynthesis of secoiridoids in Gentiana rhodantha, which could be useful in the cultivation of Gentiana rhodantha. However, the data should be better analyzed, the result should be better presented, and the writing should be significantly improved. To improve the quality of this manuscript, and to ensure the results are reliable and reproducible, there are some concerns to be addressed.

1. Please change the page orientation from landscape to portrait.
2. Line 36-38, “At present, …, Lianlong capsule”, it would be helpful if the authors can introduce the medical function of these compound formulations.
3. Line 38-41. “Recently, …, have not been reported yet”, it would be helpful if the authors can elaborate more about the findings of the previous studies, instead of just listing the key words for each publication.
4. Please include a figure showing representative images of different Gentiana rhodantha tissues used in this study.
5. In Fig. 5, the authors need describe the meaning of the x-axis and y-axis in details.
6. Increase the font of all figures, especially Fig. 8.

Experimental design

1. Please provide sufficient details in the materials and methods section.
(1) Please specify the amount of tissue used for RNA extraction.
(2) How was the abundance of the transcripts determined?
(3) It is “DESeq2”, not “DESeq”.
(4) What was the FDR cutoff in DESeq2? If the authors used default parameters of DESeq2, the FDR should be 10%.
(5) Please include the method for generating the heatmap of Fig. 1.
(6) For SSR analysis, the authors need to describe what was the input and what was the output.
2. In Fig. 7, the title has an error. Also, the DEG analyses need two groups, the experimental group and the control group. “Flower” or other tissue is just one group. This figure should be removed from the manuscript.

Validity of the findings

1. First, with the lack of vital details in the methods (especially the transcript abundance calling and DEG analysis), the validity of the findings cannot be fully assessed in the current form. But I will be willing to assess the validity of the findings in the next submission, if the necessary information is provided.
2. The results do not fully support the conclusion that the genes involved in secoiridoid biosynthesis pathway were “determined”. Please either reword the conclusion or present the results in a better way to support such claim.

Additional comments

1. Please make sure the institution name was provided correctly both in the manuscript and in the NCBI SRA database.
2. Please italicize “de novo”, “in vivo”.
3. Please proofread the manuscript carefully to correct any grammatical errors, typos and formatting errors. For example:
(1) there should be a space before and after the parenthesis ().
(2) Correct the grammar of Line 66-68 “Fresh roots, …, loganic acid”.
(3) Correct the grammar of Line 89-97, Line 100-101.
There are many additional errors, please try to correct them all.

---

## Round 0.2 · Minor Revisions

Please address the comments carefully.

·

Basic reporting

This is the second version of the article after Minor revisions. The authors have addressed most of the pressing concerns satisfactorily.

Experimental design

The experimental design remains the same as the original version, hence no new comments.

Validity of the findings

The findings are same as reported and reviewed in original version; hence no new comments

Additional comments

No additional comments

Reviewer 3 ·

Basic reporting

In the revised version of the manuscript titled “Transcriptome analysis and exploration of genes involved in the biosynthesis of secoiridoids in Gentiana rhodantha”, the authors made some efforts to address the reviewers’ concerns to improve the quality of this manuscript. However, there are still some major concerns need to be addressed before the publication of this manuscript.
1. As pointed out by the reviewers previously, there are some minor, but too many, copy-editing errors in this manuscript, including typographical errors, grammatical errors, punctuation errors, spacing errors, etc. The authors corrected some of these errors, however, there are still many minor errors in this manuscript. To increase the efficiency of the peer review process, I highly recommend the authors to perform several rounds of rigorous and careful proof-reading before resubmitting the manuscript. Please correct any additional errors, not just the ones pointed out by the reviewers.
2. Line 204, “As can be seen from the figure, there is a high degree of similarity between all the samples”. Obviously, this statement did not accurately describe the result shown in Figure 2.
3. The authors should perform principal component analysis for the RNA-seq data, to show the differences and similarities among samples.
4. Some of the figure titles and figure legends are too brief, and do not sufficiently describe what is shown in the figure. For example, please accurately describe the Figure 3 in sufficient details, also explain what error bars represent in this figure. Did the authors plot the number of TF in different tissues? If so, the authors should plot different tissues in separate panels, instead of putting in one plot with error bars. If not, please also explain this figure in sufficient details.
5. I requested the authors to explain the x-axis labels of Figure 5 in the previous round of review. The authors replied, “We see that Figure 5 is a pie chart”. The Figure 5 was a bar chart in the previous submission (74580-v0), which was changed to Figure 6 in the revised manuscript. I understand that the authors could generate multiple versions of manuscript and the authors might get confused with the figure numbers. However, when checking the line numbers and figure numbers in the reviewers’ comments, I recommend to authors to refer to the line numbers and figure numbers in the version submitted for review previously, not any other internal versions.
6. Line 269, it should be “DESeq2”.

Experimental design

1. The reviewer specifically requested the authors to describe the method used for transcript abundance determination in the RNA-seq analysis (see point 1 (2) of experimental design in previous review). This information is still not complete. For the article to be FAIR, please describe which package was used to map the reads to the assembled transcriptome and which packages were used to determine the transcript abundance, and describe the parameters as well.
2. The authors need to describe the methods for generating the heatmap shown in Figure 8 B-D, and describe what values were used to construct the heatmap. Please note that since the authors are comparing between samples, z-score of TPM values should be used, instead of z-score of FPKM values.

Validity of the findings

1. Since this is a de novo assembled transcriptome, the authors should evaluate the completeness of the transcriptome assembly. The authors can use BUSCO (https://busco.ezlab.org) to evaluate the completeness and report the percentage of complete, fragmented, and missing BUSCOs.
Here is the original publication of BUSCO: Simão, F. A., Waterhouse, R. M., Ioannidis, P., Kriventseva, E. V., & Zdobnov, E. M. (2015). BUSCO: assessing genome assembly and annotation completeness with single-copy orthologs. Bioinformatics, 31(19), 3210-3212.
The authors can check some publications using BUSCO to evaluate the completeness of the de novo assembled transcriptome. For example, Geng, Y., Cai, C., McAdam, S. A., Banks, J. A., Wisecaver, J. H., & Zhou, Y. (2021). A de novo transcriptome assembly of Ceratopteris richardii provides insights into the evolutionary dynamics of complex gene families in land plants. Genome biology and evolution, 13(3), evab042. This publication is just an example to demonstrate how to use BUSCO to evaluate and report the completeness of de novo assembled plant transcriptome. The authors do not need to cite this publication.

---

## Round 0.3 · Minor Revisions

Please address all the comments carefully so that there is no further delay

Reviewer 3 ·

Basic reporting

In the revised version of the manuscript titled “Transcriptome analysis and exploration of genes involved in the biosynthesis of secoiridoids in Gentiana rhodantha” (#74580-v2), the authors made additional efforts to address the reviewers’ concerns to improve the quality, reproducibility, and validity of this study. Most of the concerns were properly addressed in this revision.
1. The authors significantly improved the writing. Most of the minor errors were corrected. But there are still some remaining errors. For example:
a) Line 449-456, all the “fifinal” should be “final”.
b) Line 170, “v1. 6.3” should be “v1.6.3”. There was an extra space.
It would be helpful if the authors could proofread the manuscripts multiple times to correct any remaining errors.
2. In the previous review, the reviewer raised a concern regarding the accurate description and interpretation of the Pearson’s Correlation Coefficient result in Fig 2B. The authors have now modified the result description to “The figure shows that the correlation of three repeats within both the stem and root was > 0.8, which indicated high reproducibility. However, the correlation of three repeats within both leaf and flower was not good.” This is an improvement from the previous version. The authors properly acknowledged that there are variations in the replicates of the leaf and flower samples. It is common and acceptable to have variations in biological replicates, but it is not acceptable to ignore the variation and claim they are well-replicated, as what was written in the previous versions.
3. The authors took the reviewer’s suggestion to perform the BUSCO analysis to evaluate the completeness of the de novo assembled transcriptome. The assembly quality is good.
a) However, in the result section, the authors wrote “we recovered 1 614 of the 1 292 conserved eukaryotic genes (80%)” (line 212-213). Should it be “we recovered 1 292 of the 1 614 conserved eukaryotic genes (80%)”?
b) The authors should specify the version of BUSCO they used.
4. Since the authors now have a transcriptome assembly with good quality, the authors should make the assembled transcriptome (FASTA file) publicly available, for example, as a supplementary file for the manuscript.
5. The authors provided more details about the transcriptome assembly and transcript abundance determination, as suggested by the reviewers. However, there are still some minor concerns.
a) It is Trinity, not “trinitf” (line 116).
b) It is “De Brujin graph”, not “De Brujin diagram” (line 119). I understand that graph and diagram are very similar words and could be confusing, but the “De Brujin graph” method is widely used in de novo sequence assembly. It is a specific mathematical term that should not be altered.
c) The authors should specify the version of all the bioinformatics packages used.
d) It is a bit confusing whether the authors used FPKM or TPM values. The authors claimed they used FPKM value in the methods section, while claimed they used TPM value in the results section. RSEM can report both FPKM value and TPM value. Please accurately describe what was used for further comparison. Again, as suggested by the reviewer in the last round, the TPM value should be used to compare between samples.
6. In the previous review, the reviewer specifically suggested the authors to “describe the methods for generating the heatmap shown in Figure 8 B-D” (See point 2 of experimental design in the previous review). However, the methods for generating the heatmap were still not described in the methods section nor in the rebuttal letter. The authors should describe what software / packages were used.
Overall, the manuscript now comes in a much better shape comparing to the previous versions, with better readability and proper data interpretation. But these remaining minor concerns still need to be addressed.

Experimental design

See above.

Validity of the findings

No further comments.

---

## Round 0.4 · accepted · Accept

The comments raised by the reviewers are addressed